# OpenReview forum: "Non-targeted Adversarial Attacks on Vision-Language Models via Maximizing Information Entropy"
_ICLR.cc/2024/Conference — Submitted to ICLR 2024_

### Official Review · Reviewer_fBK2 · 2023-10-26

**Soundness:** 2 fair
**Presentation:** 4 excellent
**Contribution:** 2 fair
**Rating:** 5
**Confidence:** 4

**Summary:**

This work proposes a novel untargeted attack on large visual-language models. To achieve this, the paper attempts to perturb the input image in order to maximize the entropy of the logits, attention, and the intermediate embeddings. Subsequently, the semantic meaning of the output texts is disrupted. The experiments demonstrate the effectiveness of this method and show that visual-language models can be attacked from the vision side.

**Strengths:**

- This paper explores an important problem. Currently, a lot of work on adversarial examples has been done on vision models, while less attention has been paid to large VLMs. With the increasing application of large VLMs, I believe that the exploration of the robustness of large VLMs is a valuable and important step.
- This paper presents the first untargeted attack against large VLMs. The untargeted goal is realized by maximizing the entropy of the intermediate or output values of the LLM.
- This paper is well-written and well-presented. I enjoyed reading this paper.

**Weaknesses:**

- While I like the idea of this paper, the technical contributions concern me. Since this paper is not the first attack against large VLMs, I believe the main contribution of this paper is the design of an untargeted attack that explores information entropy. If this is the case, I think the contributions, in their current state, lack depth. I believe this paper can improve in the following directions.
    - For example, we now have three methods of MIE, but which one is better? What are their strength and weaknesses? Can we have an analysis and an in-depth discussion?
    - What factors can influence the performance of this method?
    - Is this attack robust against simple defensive methods like robust training?
    - Comprehensive evaluation on parameter different settings. For now the evaluation is limited, e.g., only one attack strength is evaluated.

**Questions:**

I generally appreciate the direction and idea of this paper. However, the current state of this paper can only be considered a simple proof of concept and lacks depth. I would consider accepting this paper if the authors provide a comprehensive study of this method.

---

> ### Author Response · Authors · 2023-11-21
> **Analysis of our attack method**
>
> **Considering the response exceeds 5,000 characters, we will submit it in two parts.**
>
> # Part 1
> Thank you for recognizing the value of our research topic and the quality of our writing, which encourages us to continue studying this issue. In response to your questions, we provide the following analysis:
>
> 1. **Analysis of our attack method.** By inducing the information entropy of the model's logits, attention scores, and hidden states to gradually increase, we can force the model to output texts that do not conform to reality. This is because information entropy is an indicator of the uncertainty of information content. When information entropy increases, it signifies that the model's uncertainty regarding prediction results also increases, thereby reducing the logical coherence of the output text.
>      - Implementing a joint attack with different weights for these three losses offers the following advantages:
>        - **Multi-angle attack**: By attacking logits, attention scores, and hidden states simultaneously, the model's prediction process can be disrupted from multiple angles. Consequently, even if the model exhibits strong robustness in a particular aspect, it would struggle to withstand attacks from various directions.
>        - **Weight adjustment**: By adjusting the weights of different losses, optimization can be achieved according to the specific characteristics and attack objectives of the model. For instance, if the model is sensitive to logits perturbation, the weight of logits loss can be increased to improve the attack's effectiveness.
>        - **Interactive influence**: Logits, attention scores, and hidden states interact with each other within the model. A joint attack can exploit this interaction to enhance the attack's effectiveness. For example, by increasing the entropy of attention scores, the model's focus can be dispersed during the prediction process, thereby affecting the computation of hidden states and logits and reducing the logical coherence of the output text.
>        - **Stronger attack performance**: Compared to individual attacks, a joint attack can achieve higher attack effectiveness in a shorter time, as it operates simultaneously in multiple directions, improving attack efficiency.
>      - All three methods are relatively effective, with logits-based attacks appearing to be the best from the perspective of attack success rate. However, from the standpoint of adversarial training, models may find it more challenging to defend against joint attacks. The advantages and disadvantages of executing these three methods individually are as follows:
>        - **Logits-based:**
>
>          **Advantages**: Directly affects the model's output layer, enabling rapid alteration of the model's prediction results within a short time. For some models, this attack method may yield better results.
>          Disadvantages: Targets only the output layer, potentially failing to fully exploit other information within the model. Additionally, if the model possesses strong robustness, larger perturbations may be required to achieve the desired attack effect.
>        - **Attention-based:**
>
>          **Advantages**: By altering the model's attention distribution, the model can be forced to focus on irrelevant or insignificant information, thus reducing the logical coherence of the output text. This method leverages the model's internal attention mechanism, making the attack more targeted.
>          Disadvantages: The attack's effectiveness may be influenced by the model's structure and parameter settings. For some models, altering the attention distribution may not be sufficient to produce significant attack effects.
>        - **Hidden states-based:**
>
>          **Advantages**: Directly affects the model's internal representations, allowing for deeper disruption of the model's prediction process. This method may yield substantial attack effects with relatively small perturbations.
>          Disadvantages: The attack's effectiveness may be influenced by the model's complexity and robustness. Furthermore, due to the highly abstract nature of the model's internal representations, determining an effective attack strategy may be challenging.

---

> ### Author Response · Authors · 2023-11-21
> **Factors affecting the performance of MIE & Effectiveness of adversarial training**
>
> # Part 2
> 2. **Factors affecting the performance of MIE.** The main factors affecting this method include perturbation size, the number of iterations, and different loss ratio values. The ablation experiment results for this part are as follows:
>      - **Perturbation size**: As the perturbation increases, the attack effect becomes stronger. However, when the perturbation exceeds 8, the improvement in attack effect becomes weaker.
>      - **Steps of iterations**: As the number of iterations increases, the attack effect gradually strengthens. As shown in Appendix A.1, with an increased number of iterations, the model may even generate meaningless descriptions.
>      - **Different loss ratios**: On Blip, with $\lambda_1$ fixed at 0.5 and varying $\lambda_2,\lambda_3$, the results can be visualized using a heatmap, which reveals that the optimal resutls are concentrated around the ratio of $\lambda_1:\lambda_2:\lambda_3=0.5:0.06:0.06(\approx 8:1:1)$
>          | $\epsilon ( 1/255)$ | 1 | 2 | 4 | 8 | 16 | 32 | 64 |
>          | --- | --- | --- | --- | --- | --- | --- | --- |
>          | MIE | 28.01 | 23.83 | 20.67 | 17.80 | 17.79 | 17.77 | 17.99 |
>
>          | Steps | 5 | 10 | 20 | 30 | 40 | 50 | 60 | 70 | 80 | 90 | 100 |
>          | --- | --- | --- | --- | --- | --- | --- | --- | --- | --- | --- | --- |
>          | MIE | 21.02 | 19.92 | 18.84 | 19.31 | 18.75 | 18.64 | 18.45 | 18.28 | 18.18 | 17.95 | 17.80 |
>
>          | $\lambda_2$,$\lambda_3$ | 0.025 | 0.05 | 0.06 | 0.1 | 0.2 | 0.5 | 0.75 | 1 |
>          | --- | --- | --- | --- | --- | --- | --- | --- | --- |
>          | 0.025 | 18.85 | 18.34 | 18.99 | 17.98 | 18.39 | 19.09 | 19.60 | 19.65 |
>          | 0.05 | 18.90 | 18.68 | 18.35 | 19.13 | 18.06 | 18.59 | 19.30 | 19.79 |
>          | 0.06 | 18.99 | 18.07 |**17.75** | 18.25 | 18.22 | 18.25 | 18.67 | 19.14 |
>          | 0.1 | 19.53 | 18.99 | 18.18 | 18.10 | 18.37 | 18.70 | 18.98 | 19.48 |
>          | 0.2 | 19.02 | 18.63 | 18.36 | 18.87 | 18.97 | 18.48 | 18.56 | 19.29 |
>          | 0.5 | 20.35 | 18.95 | 18.78 | 18.24 | 18.24 | 18.80 | 19.03 | 20.67 |
>          | 0.75 | 20.15 | 19.22 | 19.44 | 19.37 | 19.33 | 19.25 | 19.29 | 20.76 |
>          | 1 | 20.86 | 20.30 | 20.25 | 20.69 | 20.26 | 20.19 | 20.78 | 20.00 |
> 3. **Effectiveness of adversarial training.** Following conventional adversarial training, we find that large VLMs do not exhibit significant adversarial robustness for unseen samples. Our MIE attack based on autoregressive generation of pseudo-labels has a strong attack capability. We also compared it with some other methods, inlcuding  Carlini 2023 (performing targeted attacks using random targets), Schlarmann 2023 (utilizing descriptions of clean samples as ground-truth labels), and Nayyer 2022 ( a GAN-based method). Note that these attacks did not use the exact same experimental settings. The results across different models also show that the adversarial robustness of existing large VLMs is generally weak. In the future, we will delve into the effectiveness of adversarial training for VLMs.
>       | Model | Clean | Carlini2023[1] | Schlarmann2023[2] | Nayyer2022[3] | MIE |
>       | --- | --- | --- | --- | --- | --- |
>       | BLIP | 29.79 | 20.53 | 19.87 | 24.36 | **17.80**|
>       | BLIP-2 | 30.72 | 24.58 | 24.06 | 27.78 | **21.39** |
>       | InstructBLIP | 31.36 | 24.31 | 23.80 | 25.32 | **21.65** |
>       | LLaVA | 31.52 | 24.78 | 24.12 | 25.79 | **21.41** |
>       | MiniGPT-4 | 31.44 | 24.97 | 23.16 | 24.12 | **21.11** |
> In summary, our proposed MIE method can be applied to various Transformer-based VLMs. With its ease of use and effectiveness, it can serve as a benchmark for evaluating VLMs' robutness.
>
> **The relevant experimental analysis will be updated in the paper.**
>
> [1] Carlini, Nicholas, et al. "Are aligned neural networks adversarially aligned?." arXiv preprint arXiv:2306.15447 (2023).
>
> [2] Schlarmann, Christian, and Matthias Hein. "On the adversarial robustness of multi-modal foundation models." Proceedings of the IEEE/CVF International Conference on Computer Vision. 2023.
>
> [3] Aafaq, Nayyer, et al. "Language model agnostic gray-box adversarial attack on image captioning." IEEE Transactions on Information Forensics and Security 18 (2022): 626-638.

---

> ### Author Response · Authors · 2023-11-23
> **Gentle reminder of the author-reviewer discussion deadline**
>
> We sincerely thank you for your time and effort in reviewing our paper and providing constructive comments. We have carefully read and addressed your feedback. The discussion phase is nearing its end, but we have not yet received your response. We would be more than happy to engage further if you have any additional questions or suggestions.

---

### Official Review · Reviewer_CGXF · 2023-10-29

**Soundness:** 2 fair
**Presentation:** 3 good
**Contribution:** 1 poor
**Rating:** 5
**Confidence:** 4

**Summary:**

The study achieves non-targeted attacks on VLMs by maximizing the entropy of network output, attention, and features.

**Strengths:**

* The content of the paper is very easy to understand.

**Weaknesses:**

* Lack of comparative methods. Many methods have already been proposed for attacking VLMs[1], but the authors did not compare with these methods during the experimental phase, choosing only simple Gaussian noise for comparison.

* The approach in the paper is somewhat ad-hoc. There are various ways to disrupt the expressions in network layers, such as maximizing the norm of mid-layer features. Why did the authors choose to maximize entropy for the attack? The rationale behind this was not clarified in the paper.

* Absence of ablation studies. The final attack method in the paper is composed of three losses, but the authors did not discuss the impact of different loss coefficients on the results within the article.

* There is a typo below Equation (5), where the second instance of $\lambda_1$ should be $\lambda_3$.

[1] Zhao, Yunqing, et al. "On evaluating adversarial robustness of large vision-language models." arXiv preprint arXiv:2305.16934 (2023).

**Questions:**

Please see weaknesses

---

> ### Author Response · Authors · 2023-11-21
> **Comparison with relevant and feasible works & Reason for choosing maximizing entropy as the attack method & Ablation study on loss coefficients**
>
> Thank you for your thorough reading of our paper. In response to your concerns, we provide the following analysis.
>
> 1. **Comparison with relevant and feasible works.** We have added some comparative experiments, including Carlini 2023 (performing targeted attacks using random targets), Schlarmann 2023 (utilizing descriptions of clean samples as ground-truth labels), and Nayyer 2022 ( a GAN-based method). The results show that our MIE method is more effective, taking full advantage of the non-targeted nature to increase the perturbation space. As for Yunqing2023 [4], it is a targeted black-box attack with strong dependency on the target text, which is significantly different from our experimental setting; thus, we dot not include it in the comparison.
>    | Model | Clean | Carlini2023[1] | Schlarmann2023[2] | Nayyer2022[3] | MIE |
>    | --- | --- | --- | --- | --- | --- |
>    | BLIP | 29.79 | 20.53 | 19.87 | 24.36 | **17.80**|
>    | BLIP-2 | 30.72 | 24.58 | 24.06 | 27.78 | **21.39** |
>    | InstructBLIP | 31.36 | 24.31 | 23.80 | 25.32 | **21.65** |
>    | LLaVA | 31.52 | 24.78 | 24.12 | 25.79 | **21.41** |
>    | MiniGPT-4 | 31.44 | 24.97 | 23.16 | 24.12 | **21.11** |
>
> 2. **Reason for choosing maximizing entropy as the attack method.**  We would like to emphasize that the reason for selecting the maximization of entropy as the optimization objective is to make the model's understanding of the input image more confusing. By maximizing information entropy, we are actually pursuing the maximum confusion in the model's internal information during answer generation, thereby increasing the model's uncertainty. Although there are indeed multiple ways to calculate loss, such as maximizing the norm, not all of these methods are directly related to our attack starting point. For example, the focus of maximizing the norm is on the overall magnitude of the model's internal outputs rather than uncertainty. Although in some cases, maximizing the norm may lead to increased uncertainty in the model's internal information, this connection is not as direct and clear as with entropy. We have added experiments based on the norm, and the results show that MIE has a stronger attack effect.
>    | Model | Clean | Norm-based | MIE |
>    | --- | --- | --- | --- |
>    | BLIP | 29.79 | 20.81 | **17.80** |
>    | BLIP-2 | 30.72 | 22.97 | **21.39** |
>    | InstructBLIP | 31.36 | 23.74 | **21.65** |
>    | LLaVA | 31.52 | 24.53 | **21.41** |
>    | MiniGPT-4 | 31.44 | 23.65 | **21.11** |
>
> 3. **Ablation study on loss coefficients.** Table 1 in the paper shows the results of individual attacks in the last four columns. For the BLIP model, we conduct ablation experiments by fixing $\lambda_1=0.5$ and varying $\lambda_2$ and $\lambda_3$. The results can be visualized using a heatmap, which reveals that the optimal resutls are concentrated around the ratio of $\lambda_1:\lambda_2:\lambda_3=0.5:0.06:0.06(\approx 8:1:1)$. **Additional ablation experiments can be found in Reviewer 4's comments.**
>    | $\lambda_2$,$\lambda_3$ | 0.025 | 0.05 | 0.06 | 0.1 | 0.2 | 0.5 | 0.75 | 1 |
>    | --- | --- | --- | --- | --- | --- | --- | --- | --- |
>    | 0.025 | 18.85 | 18.34 | 18.99 | 17.98 | 18.39 | 19.09 | 19.60 | 19.65 |
>    | 0.05 | 18.90 | 18.68 | 18.35 | 19.13 | 18.06 | 18.59 | 19.30 | 19.79 |
>    | 0.06 | 18.99 | 18.07 |**17.75** | 18.25 | 18.22 | 18.25 | 18.67 | 19.14 |
>    | 0.1 | 19.53 | 18.99 | 18.18 | 18.10 | 18.37 | 18.70 | 18.98 | 19.48 |
>    | 0.2 | 19.02 | 18.63 | 18.36 | 18.87 | 18.97 | 18.48 | 18.56 | 19.29 |
>    | 0.5 | 20.35 | 18.95 | 18.78 | 18.24 | 18.24 | 18.80 | 19.03 | 20.67 |
>    | 0.75 | 20.15 | 19.22 | 19.44 | 19.37 | 19.33 | 19.25 | 19.29 | 20.76 |
>    | 1 | 20.86 | 20.30 | 20.25 | 20.69 | 20.26 | 20.19 | 20.78 | 20.00 |
>
> In summary, our proposed MIE method can be rapidly applied to various Transformer-based vision-language models. Owing to its usability and effectiveness, it can serve as a new benchmark for evaluating the robutness of VLMs.
>
> **The relevant experimental analysis will be updated in the paper.**
>
> [1] Carlini, Nicholas, et al. "Are aligned neural networks adversarially aligned?." arXiv preprint arXiv:2306.15447 (2023).
>
> [2] Schlarmann, Christian, and Matthias Hein. "On the adversarial robustness of multi-modal foundation models." Proceedings of the IEEE/CVF International Conference on Computer Vision. 2023.
>
> [3] Aafaq, Nayyer, et al. "Language model agnostic gray-box adversarial attack on image captioning." IEEE Transactions on Information Forensics and Security 18 (2022): 626-638.
>
> [4] Zhao, Yunqing, et al. "On evaluating adversarial robustness of large vision-language models." arXiv preprint arXiv:2305.16934 (2023).

---

> ### Author Response · Authors · 2023-11-23
> **Gentle reminder of the author-reviewer discussion deadline**
>
> We sincerely thank you for your time and effort in reviewing our paper and providing constructive comments. We have carefully read and addressed your feedback. The discussion phase is nearing its end, but we have not yet received your response. We would be more than happy to engage further if you have any additional questions or suggestions.

---

### Official Review · Reviewer_qFBU · 2023-11-01

**Soundness:** 3 good
**Presentation:** 3 good
**Contribution:** 2 fair
**Rating:** 6
**Confidence:** 3

**Summary:**

The paper proposes a new attack on VLM. They utilize the PGD optimization algorithm to maximize the entropy of the predicted token entropy, attention weights, and normalized hidden layers values. They show empirically that their attack is effective in attacking sevral open-source VLMs.

**Strengths:**

- The topic of robustness of VLMs is relevant.
- The experimental results show the effectiveness of the attack.

**Weaknesses:**

- The technical contribution is on the moderate side. However, to the best of my knowledge using maximum entropy to adversarial attack is novel.
- the paper lacks comparison to related work. There are several attacks on image captioning in the literature. The paper lacks a comparison against them (see for example, [1] and [2]). The authors mention [2] in the related work but did not empirically compare against it, they justify it as this method uses the original caption. This, however, is a minor requirement as it can be mitigated by treating the caption on the clean image as the ground-truth caption.


[1] Aafaq, Nayyer, et al. "Language model agnostic gray-box adversarial attack on image captioning." IEEE Transactions on Information Forensics and Security 18 (2022): 626-638.

[2] Schlarmann, Christian, and Matthias Hein. "On the adversarial robustness of multi-modal foundation models." Proceedings of the IEEE/CVF International Conference on Computer Vision. 2023.

**Questions:**

- There are hyper-parameters for the attacks $\lambda_1,\lambda_2,\lambda_3$. Are these selected based on the same validation set?  If so I believe the experiment might lack statistical integrity.

---

> ### Author Response · Authors · 2023-11-21
> **Technical contributions & Comparison with relevant and feasible works & Selection of the validation set**
>
> Thank you for acknowledging the novelty of our attack method. In response to your concerns, we provide the following analysis.
>
> 1. **Technical contributions.** Existing attacks on large Very Large Models (VLMs) generally adopt similar approaches, utilizing teacher forcing to calculate the loss of target texts. These methods are relatively simple but demonstrate a certain level of effectiveness. Our method based on maximizing information entropy provides a new attack perspective and is highly effective. Precisely because the technical implementation of MIE is relatively straightforward, it can encourage more researchers to focus on the adversarial robustness of VLMs.
>
> 2. **Comparison with relevant and feasible works.** Below are our comparative experimental results inlcuding  Carlini 2023 (performing targeted attacks using random targets), Schlarmann 2023 (utilizing descriptions of clean samples as ground-truth labels), and Nayyer 2022 ( a GAN-based method).:
>    | Model | Clean | Carlini2023[1] | Schlarmann2023[2] | Nayyer2022[3] | MIE |
>    | --- | --- | --- | --- | --- | --- |
>    | BLIP | 29.79 | 20.53 | 19.87 | 24.36 | **17.80**|
>    | BLIP-2 | 30.72 | 24.58 | 24.06 | 27.78 | **21.39** |
>    | InstructBLIP | 31.36 | 24.31 | 23.80 | 25.32 | **21.65** |
>    | LLaVA | 31.52 | 24.78 | 24.12 | 25.79 | **21.41** |
>    | MiniGPT-4 | 31.44 | 24.97 | 23.16 | 24.12 | **21.11** |
>
> 3. **Selection of the validation set.** Our validation set comprises 1,000 randomly selected images, which inherently provide a certain level of generalization. This selection method is employed to maintain consistency with related works in the adversarial attack domain. Furthermore, we conduct ablation experiments by fixing $\lambda_1=0.5$ and varying $\lambda_2$ and $\lambda_3$ for the BLIP model. The results can be visualized using a heatmap, which reveals that the optimal resutls are concentrated around the ratio of $\lambda_1:\lambda_2:\lambda_3=0.5:0.06:0.06(\approx 8:1:1)$. **Additional ablation experiments can be found in Reviewer 4's comments.**
>    | $\lambda_2$,$\lambda_3$ | 0.025 | 0.05 | 0.06 | 0.1 | 0.2 | 0.5 | 0.75 | 1 |
>    | --- | --- | --- | --- | --- | --- | --- | --- | --- |
>    | 0.025 | 18.85 | 18.34 | 18.99 | 17.98 | 18.39 | 19.09 | 19.60 | 19.65 |
>    | 0.05 | 18.90 | 18.68 | 18.35 | 19.13 | 18.06 | 18.59 | 19.30 | 19.79 |
>    | 0.06 | 18.99 | 18.07 |**17.75** | 18.25 | 18.22 | 18.25 | 18.67 | 19.14 |
>    | 0.1 | 19.53 | 18.99 | 18.18 | 18.10 | 18.37 | 18.70 | 18.98 | 19.48 |
>    | 0.2 | 19.02 | 18.63 | 18.36 | 18.87 | 18.97 | 18.48 | 18.56 | 19.29 |
>    | 0.5 | 20.35 | 18.95 | 18.78 | 18.24 | 18.24 | 18.80 | 19.03 | 20.67 |
>    | 0.75 | 20.15 | 19.22 | 19.44 | 19.37 | 19.33 | 19.25 | 19.29 | 20.76 |
>    | 1 | 20.86 | 20.30 | 20.25 | 20.69 | 20.26 | 20.19 | 20.78 | 20.00 |
>
> In conclusion, given the usability and effectiveness of our proposed MIE method, it can serve as a new benchmark for evaluating the robustness of VLMs.
>
> **The relevant experimental analysis will be updated in the paper.**
>
> [1] Carlini, Nicholas, et al. "Are aligned neural networks adversarially aligned?." arXiv preprint arXiv:2306.15447 (2023).
>
> [2] Schlarmann, Christian, and Matthias Hein. "On the adversarial robustness of multi-modal foundation models." Proceedings of the IEEE/CVF International Conference on Computer Vision. 2023.
>
> [3] Aafaq, Nayyer, et al. "Language model agnostic gray-box adversarial attack on image captioning." IEEE Transactions on Information Forensics and Security 18 (2022): 626-638.

---

> > ### Comment · Reviewer_qFBU · 2023-11-22
> > **Response**
> >
> > Thank you for you response.
> >
> > I appreciate the comparison study, It improves the paper significantly. However, the selection of the hyper-parameters on the same test set is flawed. These parameters should be selected on an independent validation set to preserve statistical integrity of the experiment. Therefore, I can only increase my score by only one.

---

> > > ### Author Response · Authors · 2023-11-23
> > > **hyper-parameters experiments on the COCO val 2017 dataset**
> > >
> > > We greatly appreciate your increased score. In response to your remaining concerns, we have supplemented our experiments on the COCO val 2017 dataset. The experimental results demonstrate that our method still maintains excellent performance and exhibits a similar patter, with the best results occurring around  the ratio of $\lambda_1:\lambda_2:\lambda_3=0.5:0.06:0.06(\approx 8:1:1)$.
> > >    | $\lambda_2$,$\lambda_3$ | 0.025 | 0.05 | 0.06 | 0.1 | 0.2 | 0.5 | 0.75 | 1 |
> > >    | --- | --- | --- | --- | --- | --- | --- | --- | --- |
> > >    | 0.025 | 18.46 | 18.31 | 18.21 | 18.97 | 18.68 | 19.25 | 19.77 | 19.85 |
> > >    | 0.05 | 18.43 | 18.65 | 17.97 | 18.35 | 18.34 | 18.89 | 19.65 | 19.87 |
> > >    | 0.06 | 18.65 | 18.24 |**17.69** | 18.17 | 18.36 | 18.94 | 19.28 | 19.24 |
> > >    | 0.1 | 18.70 | 18.86 | 18.06 | 18.15 | 18.23 | 18.88 | 19.31 | 19.32 |
> > >    | 0.2 | 18.89 | 19.50 | 18.38 | 18.69 | 18.65 | 18.76 | 19.24 | 19.58 |
> > >    | 0.5 | 19.01 | 19.96 | 19.20 | 18.57 | 18.31 | 18.65 | 19.26 | 20.15 |
> > >    | 0.75 | 19.32 | 20.11 | 19.46 | 19.50 | 19.39 | 19.32 | 19.45 | 20.03 |
> > >    | 1 | 19.65 | 20.17 | 20.25 | 19.98 | 20.17 | 19.45 | 20.02 | 19.45 |
> > >
> > >
> > > We believe that our method possesses strong generalization capabilities and an extremely high attack success rate, which indicates an urgent need for adversarial robustness research in current vision-language models.

---

### Official Review · Reviewer_r3mc · 2023-11-06

**Soundness:** 2 fair
**Presentation:** 2 fair
**Contribution:** 2 fair
**Rating:** 5
**Confidence:** 5

**Summary:**

The paper proposes  adversarial attacks on Visual Language Models (VLMs), as illustrated in Figure 1. A clean image is perturbed slightly so that a VLM generates a wrong description of the image. In this paper, the attacks are "non-targeted" in that the goal is to cause the description to change to anything, and not something specific.

A concern with the paper is that prior work, e.g., (Carlini 2023), which is cited in the paper, also solves a pretty similar problem, but the contribution over that work  is not clearly stated. For instance, see page 20 of (Carlini 2023) in which the image of Mona Lisa was adversarially perturbed to cause a completely incorrect description to be output. Why couldn't the same techniques be used here and what precisely is the contribution over (Carlini 2023) or similar work? I would have liked to see the difference with the closest works highlighted in the Intro.

In the Related Work section, it seems that the main difference claimed is that prior adversarial attacks on VLMs are targeted, whereas authors propose an untargeted attack (as an aside, if this is the crucial contribution -- I think that should have been stated in the Intro clearly). But, even accepting the author's premise that untargeted attacks weren't addressed by prior work, why (1) is an untargeted attack important; (2) not a special case of a targeted attack where a target is picked at random from the desired domain, thus turning an untargeted attack into a targeted attack and using a prior solution.

**Strengths:**

The specific algorithm for doing an untargeted attack seems to be different from prior work in VLMs. The authors propose injecting adversarial noise that maximizes the entropy  -- thus effectively causing the resulting image to produce essentially a random caption. They propose three different ways of maximizing entropy and use a weighted combination of the three methods as the objective function (as shown in Algorithm 1).  They find that the first method (equation 2) dominates overall (weight ended up as 0.8). An ablation study would  have been nice to show how each way (equation 2, 3, or 4) would have performed by itself versus the  weighted combination of the three.

**Weaknesses:**

An alternate way would be to simply feed some random image that is clearly maximum entropy to the VLM, get a few words of text from it (essentially a random phrase) and then use a targeted attack on the VLM to generate that particular or similar text. I would have liked to see a comparison with such an approach in the paper.

As an example, couldn't "a pair of flip flops sitting on a pile of garbage" in Appendix A.1 first row  be set as the target caption in Carlini 2023 and then a perturbation found that achieves that? Or any other random caption for that matter?

Wouldn't the captions just become random sequence of words as the attack progresses? The figures in Appendix A should probably illustrate that, if that is the case. And if that is the case, is such an attack considered successful? Or should the caption generally make sense to a human?

**Questions:**

Can the Intro be revised to identify the closest work to the paper and identify the key contribution over that work (or minor variants of prior work)?

Why can't untargeted attacks use a method for targeted attacks as a subroutine to achieve an untargeted attack? Note that there is precedence for this in adversarial ML. In the OPT method for blackbox adversarial attacks, the fundamental method is for a targeted attack. The OPT paper discusses how to do an untargeted attack by wrapping a small amount of code around a targeted attack. An example strategy would be to choose a random target and then attempt a targeted attack.

Why are untargeted attacks on VLMs particularly interesting, given that we already know that targeted attacks are possible and how to do them?

Can an ablation study be presented to show how each entropy method performs on its own (e.g., lambda_1 = 1, others 0, etc.), versus the chosen setting for lambdas of (0.8, 0.1., 0.1).

---

> ### Author Response · Authors · 2023-11-21
> **Key Contribution & Feasibility of targeted attacks using random targets**
>
> **Considering the response exceeds 5,000 characters, we will submit it in two parts.**
> # Part 1
>
> Thank you for providing insightful feedback, which is greatly helpful for improving our paper. Regarding the four questions you raised, here is our analysis:
> 1. **Key Contribution.** We propose the MIE method, which focuses on  non-targeted attacks, characterized by weaker constraints and a larger optimization space. Unlike previous methods, such as [1] and [2], which involve targeted optimization of adversarial samples towards specific directions, our approach is more suitable for image description tasks. Supplementary experiments demonstrate that the attack performance of MIE surpasses that of comparative methods. We plan to revise and upload a new version of the paper shortly. MIE is the first untargeted attack method that does not rely on the true descriptions of images, exhibiting strong generalization capabilities.
> 2. **Feasibility of targeted attacks using random targets.** While it is computationally feasible to implement non-targeted attacks using targeted attack methods as subroutines, it is insufficient for  image description tasks. Unlike traditional classification tasks, images do not have only one correct description, especially for complex images with multiple description angles. For instance, "it's a sunny day" and "the sunshine is so bright" are textually inconsistent but semantically equivalent. Therefore, the success rate of such random targeted attacks is relatively low. Below are our comparative experimental results inlcuding  Carlini 2023 (performing targeted attacks using random targets), Schlarmann 2023 (utilizing descriptions of clean samples as ground-truth labels), and Nayyer 2022 ( a GAN-based method):
>       | Model | Clean | Carlini2023[1] | Schlarmann2023[2] | Nayyer2022[3] | MIE |
>       | --- | --- | --- | --- | --- | --- |
>       | BLIP | 29.79 | 20.53 | 19.87 | 24.36 | **17.80**|
>       | BLIP-2 | 30.72 | 24.58 | 24.06 | 27.78 | **21.39** |
>       | InstructBLIP | 31.36 | 24.31 | 23.80 | 25.32 | **21.65** |
>       | LLaVA | 31.52 | 24.78 | 24.12 | 25.79 | **21.41** |
>       | MiniGPT-4 | 31.44 | 24.97 | 23.16 | 24.12 | **21.11** |

---

> ### Author Response · Authors · 2023-11-21
> **Non-targeted attacks pose weaker constraints compared to targeted attacks & Ablation study on loss coefficients**
>
> # Part 2
> 3. **Non-targeted attacks pose weaker constraints compared to targeted attacks.**
>    - Non-targeted attacks do not aim to make VLMs produce specific outputs, resulting in a larger solution space for numerical optimization. This is also a reason why targeted attacks may be less effective.
>    - MIE can serve as a powerful method for assessing the robustness of VLMs. As the attack progresses, the model's output may become nonsensical, consisting of repetitive random words. This indicates that the VLM's robustness is insufficient, as it cannot generate coherent sentences, which in turn diminishes user experience and weakens user trust. In scenarios with high security requirements, the lack of robustness can lead to severe consequences.
>    - We also experimentally found that conventional adversarial training cannot defend against our attack, indicating that current VLMs are extremely vulnerable to our attack method.
>
>
> 4. **Ablation study on loss coefficients.** Table 1 in the paper shows the results of individual attacks in the last four columns. For the BLIP model, we conduct ablation experiments by fixing $\lambda_1=0.5$ and varying $\lambda_2$ and $\lambda_3$. The results can be visualized using a heatmap, which reveals that the optimal resutls are concentrated around the ratio of $\lambda_1:\lambda_2:\lambda_3=0.5:0.06:0.06(\approx 8:1:1)$. **Additional ablation experiments can be found in Reviewer 4's comments.**
>       | $\lambda_2$,$\lambda_3$ | 0.025 | 0.05 | 0.06 | 0.1 | 0.2 | 0.5 | 0.75 | 1 |
>       | --- | --- | --- | --- | --- | --- | --- | --- | --- |
>       | 0.025 | 18.85 | 18.34 | 18.99 | 17.98 | 18.39 | 19.09 | 19.60 | 19.65 |
>       | 0.05 | 18.90 | 18.68 | 18.35 | 19.13 | 18.06 | 18.59 | 19.30 | 19.79 |
>       | 0.06 | 18.99 | 18.07 |**17.75** | 18.25 | 18.22 | 18.25 | 18.67 | 19.14 |
>       | 0.1 | 19.53 | 18.99 | 18.18 | 18.10 | 18.37 | 18.70 | 18.98 | 19.48 |
>       | 0.2 | 19.02 | 18.63 | 18.36 | 18.87 | 18.97 | 18.48 | 18.56 | 19.29 |
>       | 0.5 | 20.35 | 18.95 | 18.78 | 18.24 | 18.24 | 18.80 | 19.03 | 20.67 |
>       | 0.75 | 20.15 | 19.22 | 19.44 | 19.37 | 19.33 | 19.25 | 19.29 | 20.76 |
>       | 1 | 20.86 | 20.30 | 20.25 | 20.69 | 20.26 | 20.19 | 20.78 | 20.00 |
>
> In summary, our proposed MIE method has two main advantages:
> 1. **Usability**: The MIE method carries out attacks in an unsupervised manner, with weaker constraints and no reliance on true labels, making it more convenient to implement. This can encourage more researchers to focus on the robustness of VLMs.
> 2. **Effectiveness**: Supplementary experiments show that MIE has a stronger attack effect than existing methods, achieving almost a 100% attack success rate, and conventional adversarial training cannot defend against it.
>
> This usability and effectiveness make our method a new benchmark for evaluating the adversarial robustness of VLMs.
> **The relevant experimental analysis will be updated in the paper.**
>
> [1] Carlini, Nicholas, et al. "Are aligned neural networks adversarially aligned?." arXiv preprint arXiv:2306.15447 (2023).
>
> [2] Schlarmann, Christian, and Matthias Hein. "On the adversarial robustness of multi-modal foundation models." Proceedings of the IEEE/CVF International Conference on Computer Vision. 2023.
>
> [3] Aafaq, Nayyer, et al. "Language model agnostic gray-box adversarial attack on image captioning." IEEE Transactions on Information Forensics and Security 18 (2022): 626-638.

---

> ### Author Response · Authors · 2023-11-23
> **Gentle reminder of the author-reviewer discussion deadline**
>
> We sincerely thank you for your time and effort in reviewing our paper and providing constructive comments. We have carefully read and addressed your feedback. The discussion phase is nearing its end, but we have not yet received your response. We would be more than happy to engage further if you have any additional questions or suggestions.

---

### Meta-Review · Area_Chair_Q6Au · 2023-12-06

**Metareview:**

This paper proposes an adversarial attack to corrupt the generated caption (response) from a CLIP-style VLM, using what is called a maximum entropy loss, where the designed attack consists of three levels of loss at attention level, output level, and each layer. The resulting attack seems to be more powerful than existing attacks, such as (Carlini et al, 2023) in fooling the model to get an incorrect (and nonsensical) label. The authors engaged in an effective author/reviewer discussion which eventually led to score increases from two reviewers. However, even after the rebuttal, there are several major concerns with the paper as follows:

1. The design is quite involved, and I would have loved to see ablations that remove each of the loss components rather than hyperparameter search table. From the new tables, it seems that keeping only $\lambda_1$ may indeed lead to a model that is on par with (Carlini et al, 2023).

2. In principle the target based attacks can also be applied in the three different hierarchies defined in this paper, so comparing with the vanilla model might be unfair.

3. A simple baseline could be to just obtain a (say prompt-based or trained) classifier on the images, and then attack the classifier to perturb the images and use it within the VLM. This is a weak baseline that I think could have portrayed a better picture of the strength of the attacks.

4. Several reviewers expressed dismay at the nonsensical nature of the captions due to attacks but this is not necessarily a blocker for publication.

Overall, I think the paper falls short of acceptance given that, as it stands, there is insufficient evidence for the necessity and effectiveness of the complex design proposed herein. As such the paper is recommended to be rejected at this time. The authors are encouraged to take the reviewers' feedback into account for a future submission of their work.

**Justification For Why Not Higher Score:**

The major concern that led to the reject decision is that there is insufficient evidence for the effectiveness of the proposal in this work. The proposed method has several moving parts and it is not clear which part has led to real improvements.

**Justification For Why Not Lower Score:**

The paper is on a timely subject, and the method seems to be effective.

---

### Decision · Program_Chairs · 2024-01-16

Reject